# Lipid and Polymeric Nanoparticles: Successful Strategies for Nose-to-Brain Drug Delivery in the Treatment of Depression and Anxiety Disorders

**DOI:** 10.3390/pharmaceutics14122742

**Published:** 2022-12-08

**Authors:** Margarida Alberto, Ana Cláudia Paiva-Santos, Francisco Veiga, Patrícia C. Pires

**Affiliations:** 1Faculty of Pharmacy, University of Coimbra (FFUC), Azinhaga de Santa Comba, 3000-548 Coimbra, Portugal; 2Rede de Química e Tecnologia/Laboratório Associado para a Química Verde (REQUIMTE/LAQV), Group of Pharmaceutical Technology, Faculty of Pharmacy, University of Coimbra, 3000-548 Coimbra, Portugal; 3Health Sciences Research Centre, University of Beira Interior (CICS-UBI), Av. Infante D. Henrique, 6200-506 Covilhã, Portugal

**Keywords:** anxiety, depression, intranasal, nose-to-brain, nanoparticles

## Abstract

Intranasal administration has gained an increasing interest for brain drug delivery since it allows direct transport through neuronal pathways, which can be quite advantageous for central nervous system disorders, such as depression and anxiety. Nanoparticles have been studied as possible alternatives to conventional formulations, with the objective of improving drug bioavailability. The present work aimed to analyze the potential of intranasal nanoparticle administration for the treatment of depression and anxiety, using the analysis of several studies already performed. From the carried-out analysis, it was concluded that the use of nanoparticles allows the drug’s protection from enzymatic degradation, and the modulation of its components allows controlled drug release and enhanced drug permeation. Furthermore, the results of in vivo studies further verified these systems’ potential, with the drug reaching the brain faster and leading to increased bioavailability and, consequently, therapeutic effect. Hence, in general, the intranasal administration of nanoparticles leads to a faster onset of action, with increased and prolonged brain drug concentrations and, consequently, therapeutic effects, presenting high potential as an alternative to the currently available therapies for the treatment of depression and anxiety.

## 1. Introduction

Mental health plays an extremely important role in society, and additional investment is needed in order to not only facilitate access to effective and existing treatments but also to strengthen the research and development of new treatments [1,2].

Depression is one of the leading causes of disability worldwide, contributing to the development of other diseases, which include neuropsychiatric, metabolic, and cardiovascular conditions. It is characterized by disinterest in day-to-day activities, sadness, irritability, fatigue, feelings of guilt, low self-esteem, sleep disorders, and suicidal thoughts, among others, which is associated with a decreased quality of life. Depression can be classified into two main groups: major depression, when symptoms are more intense, and there is a greater emotional burden, or minor depression, when there are fewer symptoms, allowing daily activities to be done properly. The most accepted theory regarding the pathogenesis of depression is based on a monoaminergic transmission disorder, in other words, a disruption in the transmission of serotonin, norepinephrine and dopamine in the brain due to the complex interaction of several social (e.g., traumatic life), psychological (e.g., personality) and biological (e.g., hereditary predisposition) factors. Most drugs used in the treatment of depression increase the availability of these neurotransmitters in the synaptic cleft, using several mechanisms of action, and include: selective serotonin reuptake inhibitors, such as fluoxetine, sertraline, and escitalopram; serotonin and norepinephrine reuptake inhibitors, such as venlafaxine; tricyclic antidepressants, such as amitriptyline; non-tricyclic antidepressants, such as trazodone and mirtazapine; monoamine oxidase inhibitors (MAOIs), such as moclobemide; among others, such as agomelatine, which acts as a melatonin receptor agonist and as a selective antagonist of serotonin reuptake [3,4,5,6,7,8].

On the other hand, anxiety is a type of central nervous system (CNS) disorder that frequently interferes with day-to-day activities and decreases performance in the workplace or in school. It can also be associated with increased cardiovascular morbidity and mortality. The most frequent symptoms of anxiety include feelings such as worry, helplessness, or fear that manifest physically through increased heart and respiratory rate, sweating, and tremors. There are different types of anxiety, such as obsessive-compulsive disorder, post-traumatic stress disorder, generalized anxiety disorder, social anxiety disorder, or separation anxiety disorder. Neuronal excitability is downregulated by the synaptic transmission of γ-aminobutyric acid (GABA); thus, when this system is inhibited, symptoms associated with anxiety arise. Pharmacological treatment of anxiety includes mostly anxiolytic drugs such as benzodiazepines (e.g., diazepam or alprazolam) that bind to the GABA_A_ receptor, increasing GABAs affinity for the receptor and enhancing its action [9,10,11].

Depression and anxiety are two pathologies that are often associated; thus, a patient with depression tends to develop anxiety, and similarly, a patient with anxiety can progress to a depressive state. The presence of the two pathologies simultaneously intensifies the symptoms of both, making their treatment even more important [12,13].

Currently, the oral route of administration is the preferred route for drug administration in the treatment of depression and anxiety. However, this route has several disadvantages since drugs administered orally are exposed to first-pass hepatic metabolism and require regular administrations to ensure the constant presence of the drug at the site of action, the amount of drug that reaches the site of action (the brain) is restricted by the low permeability of the blood-brain barrier, and fluctuations in plasma concentrations can lead to side effects and loss of efficacy [7,8,14,15].

Therefore, since the prevalence of depression and anxiety has been increasing over the last few years, a growing number of studies have been developed in order to discover alternatives to the current treatments of these pathologies. One of these alternatives is the intranasal route of administration.

### 1.1. Intranasal Administration for Brain Drug Delivery

The intranasal route of administration is an alternative to the administration of drugs in the treatment of CNS pathologies which has several advantages over other routes, allowing to overcome the blood-brain barrier; transport of molecules of larger dimensions (up to about 1000 Da); avoid the first pass hepatic metabolism; minimize the side effects caused by drugs administered through the systemic circulation, reducing toxicity; reduce the drug dose that is necessary to achieve a therapeutically effective concentration at the site of action, and thus reach the therapeutic threshold. Additionally, in general, as a route of administration, it is simple, practical, and convenient since it does not require administration techniques involving coordination or swallowing (such as the oral route) or the aid of a health professional (such as the intravenous route), and noninvasive, which can contribute to increasing patient compliance [16,17,18].

These advantages are connected to the unique anatomy of the nasal cavity, which includes a direct connection to the CNS, allowing the drug to be transported to the brain. This makes the nasal cavity the only place in the human body where the nervous system is in direct contact with the surrounding environment. More specifically, the nasal cavity is divided into two parts by the nasal septum, and each part consists of three distinct regions: the vestibule, the olfactory region, and the respiratory region. The vestibule is located at the entrance of the nasal cavity and is responsible for the filtration of inhaled particles. It is the region that least contributes to the absorption of drugs. The respiratory region consists of turbinates, which are responsible for humidification and regulation of the temperature of inhaled air, contributing to the formation of an airflow that improves the contact between inhaled air and the nasal mucosa. The epithelium cells of this region are covered by microvilli and long cilia, which contribute to improved absorption. It is the main responsible region for systemic drug absorption, having a surface area of about 160 cm^2^ and high vascularization, which provides a high blood flow. From systemic circulation, the drug can be transported to the brain, but in that case, it needs to cross the blood-brain barrier in order to reach the CNS. It can also contribute to the direct absorption of drugs into the CNS through the trigeminal nerve. The olfactory region occupies a surface area of approximately 10 cm^2^, playing a crucial role in the absorption and transport of drugs to the brain. The olfactory epithelium consists of olfactory nerves, responsible for the direct transport of the drug from the nasal cavity to the brain, using the olfactory nerve. Hence, in short, intranasally administered drugs can reach the site of action in the brain directly, using the olfactory or trigeminal nerves (direct transport), or indirectly, being absorbed into the systemic circulation and then crossing the blood-brain barrier (indirect transport) (Figure 1) [16,17,19,20].

Nevertheless, in spite of the many important advantages that intranasal administration presents, it is necessary to consider some limiting factors that hinder drug absorption, namely: the physical removal of the drug from the nasal cavity by mucociliary clearance mechanisms; enzymatic degradation in the mucus and nasal epithelium layer; and the volume of formulation that can be administered, that is limited to 25–200 μL, which turns this route of administration more appropriate for potent drugs. Mucociliary clearance has the function of protecting the respiratory system against bacteria and inhaled particles as a result of the combined effect of mucus and cilia that transport the particles from the anterior to the posterior region of the nasal cavity, being eliminated to the bottom of the throat. Hence, drug bioavailability is diminished by the high flow of nasal secretions and ciliary movement since these decrease the time of residence of the drugs in the nasal cavity, affecting the permeability through the mucosa. The enzymatic activity in the nasal cavity also constitutes a barrier to the absorption of drugs since several enzymes, such as the various isoforms of cytochrome P450, are present in the nasal mucosa and perform enzymatic degradation of various drugs [17,19,20,21]. The many strategies that have been developed to overcome these obstacles, as well as the factors that influence nasal drug absorption, are addressed in the following Section 1.1.1 and Section 1.1.2.

#### 1.1.1. Physicochemical and Formulation Factors That Influence Nasal Drug Absorption

Nasal absorption is influenced by several factors, namely anatomical and physiological factors (mucociliary clearance, enzymatic degradation, membrane transport, mucosal irritation, and deposition), physicochemical factors (molecular drug weight, lipophilicity and ionization state), and formulation factors (type and characteristics of the formulation, volume of administration, drug strength, viscosity, pH, osmolarity). When the drug is administered intranasally, it comes into direct contact with the nasal mucosa. The passage through the mucus layer is the first step to its absorption. Mucin, the main protein of the mucus, binds to drugs, hampering their diffusion. Additionally, alterations in pH or temperature may alter the structure of the mucus, making the diffusion of the drug more difficult. Moreover, the mucus has elastic and viscous properties that influence the transport of drugs; hence, if the mucus is more viscous, mucociliary clearance is reduced, and, therefore, the contact time between the drug and the mucosa is increased, which may contribute to improved drug absorption [17,20,21].

After passing through the mucus layer, the transport of the drug can be performed by different mechanisms that include: transcellular diffusion, in which transport is carried out through the membrane by passive diffusion or active transport; and paracellular diffusion, a passive process, in which the drug moves through the intercellular space. The lipophilicity of the drug is one of the most important factors that determines whether the transport will be performed by transcellular diffusion or paracellular diffusion. Lipophilic drugs are preferably transported through transcellular diffusion, showing fast and efficient absorption when administered intranasally. Hydrophilic drugs are mainly transported via paracellular diffusion, resulting in low absorption. The rate of a drug’s diffusion through the nasal mucosa is also influenced by the state of ionization and molecular weight of the drug. Consequently, lipophilic and uncharged (neutral) drugs with low molecular weight are more easily absorbed when compared to hydrophilic, charged and/or high molecular weight molecules. Absorption also depends on the pKa of the drug and the pH at the absorption site, which presents values between 5.0 and 6.5 in the nasal mucosa. It is also influenced by the solubility of the drug since nose secretions have a more watery nature; therefore, the drug should present an appropriate aqueous solubility for better dissolution in the mucus itself [16,17,21,22].

In order to optimize drug transport to the action site after an intranasal administration, it is necessary to take into account certain characteristics of the formulation, for example, factors such as its capability to adhere to the mucosa, nasal permeability, and drug deposition in the olfactory epithelium. These characteristics should be elevated and should also allow a controlled and constant release of the drug. The excipients of the formulation should be selected, taking into account their functions, with the purpose of assigning properties to the formulation that allow to protect the drug and favor its administration and arrival of the drug at the action site. The excipients must also be compatible with the active substance and non-toxic or irritating to the nasal mucosa [17]. The concentration of the drug in the formulation and the volume of administration should also be taken into account since it is only possible to administer approximately 200 μL of formulation in the nasal cavity, which makes this route of administration suitable for potent drugs but challenging to drugs that require high doses or that have reduced solubility [17,18,23].

The pH of the formulation is also important and should be close to the pH of the nasal cavity to avoid mucosal irritation. Lysozyme found in nose secretions contributes to the dissolution of certain bacteria and helps to maintain acidic pH. Additionally, the pH of the formulation should be selected considering the stability of the drug and should also contribute to the existence of a higher fraction of non-ionized drug, while maintaining the functionality of the excipients. Another very important factor is the viscosity of the formulation, which should ensure contact with the nasal mucosa for an adequate period of time. Higher viscosity increases the time of contact of the drug with the mucosa and may contribute to increased absorption. However, formulations that are excessively viscous can decrease the diffusion of the drug from the formulation itself, reducing its absorption. Gel formulations can be used to increase the time of permanence of the drug in the nasal cavity and, consequently, improve its bioavailability. The most used gelling agents include cellulose derivatives (methylcellulose or carboxymethylcellulose) and carbopol. Also, considering that isotonic solutions are better tolerated, the osmolarity of the formulations should be between 285 and 310 mOsmol/L to avoid mucosal irritation. However, hypertonic solutions can be used since they transiently reduce ciliary activity, which can increase the retention time of the drug in the nasal cavity, promoting its absorption. Nevertheless, caution is required when hypertonic solutions are used in order to ensure that no damage is caused to the nasal mucosa [17,18,24,25,26,27,28,29].

#### 1.1.2. Strategies Used to Improve Drug Absorption

Many different strategies have been developed in order to improve drug solubilization and absorption, such as the use of prodrugs, absorption enhancers, enzymatic inhibitors, mucoadhesive agents, and nanometric drug transport systems [30].

Structural changes in the drug molecule can help improve their characteristics by changing physicochemical properties such as molecular weight, partition coefficient, and solubility. These can be used to improve formulation drug strength, absorption through biological barriers, or premature metabolism. The grand majority of prodrugs are administered in their inactive form, requiring biotransformation to become their active form, which produces a pharmacological effect [17,28,29,31,32,33].

Absorption enhancers are excipients that can improve drug permeability, which is especially important for hydrophilic drugs. They act by altering the phospholipid bilayer and membrane fluidity or by alternately opening tight junctions between epithelial cells, improving paracellular transport. The most commonly used are surfactants (e.g., polysorbate, poloxamers), bile salts (e.g., sodium cholate), fatty acids (e.g., stearic acid, palmitic acid), chelators (e.g., ethylenediaminetetraacetic acid (EDTA), salicylates) and polymers (e.g., chitosan, poly(D,L–lacitde co– glycolide) (PLGA)) [17,22,34,35,36].

Enzymatic inhibitors can also be used to protect drugs against the enzymatic degradation that occurs in the nasal cavity, thereby increasing the available fraction of the drug for absorption and, consequently, bioavailability. Peptidase inhibitors and proteases, such as amastatin, boroleucine, bacitracin, and puromycin may be used [22,33,37,38].

Mucoadhesive agents increase the contact time between the formulation and the nasal mucosa, also favoring the paracellular transport of hydrophilic drugs and reducing mucociliary clearance by establishing a connection between mucin and a polymer. One of the possible mechanisms adopted by the mucoadhesive systems is the absorption of water from the nasal mucosa, which leads to the swelling of the polymer, and consequent penetration into the mucus with fixation of the formulation to the nasal cavity, improving the absorption of the drug. An example is chitosan, a biocompatible and biodegradable polymer, is widely used not only for its mucoadhesive properties but also for increasing permeability and paracellular transport, through interaction with the tight junctions [17,30,39,40,41].

Finally, another very promising and widely used strategy to improve drug absorption and bioavailability is the incorporation of drugs into nanometric transport systems, discussed in detail in the next Section 1.2.

### 1.2. Nanometric Drug Transport Systems

Nanometric drug transport systems are formulations that help transport the drug to the respective action site, allowing it to modulate the time and quantity of drug released. For the fraction that does not undergo transport via the direct route (neuronal transport), nanoparticles can be transported to the brain by interaction with the blood-brain barrier, mostly via receptor-mediated transcytosis, transporter-mediated transcytosis, absorption-mediated transcytosis, and transient opening of the blood-brain barrier itself [42,43,44,45,46,47]. These systems must be biocompatible, non-toxic, and easily eliminated or biodegradable. The main categories of nanometric systems are liposomes, nanoemulsions, and polymeric or lipid nanoparticles (Figure 2) [48].

Liposomes are vesicles composed of one or more phospholipid bilayers that surround an aqueous core and are variable in size (between 20 and 1000 nm). Their composition allows the incorporation of hydrophilic (in the core) or hydrophobic (in the phospholipid bilayers) drugs that will be transported to the site of action. Nevertheless, they have some disadvantages, which include system instability, the unwanted release of the encapsulated substance, irreproducibility between batches, difficulty controlling liposome size, and low encapsulation efficiency [49,50]. Nanoemulsions are thermodynamically stable systems, consisting of two immiscible liquids (oil and water) that, when mixed, form a single phase through the action of an emulsifying agent. These systems are usually smaller than 500 nm, and can either be oil-in-water or water-in-oil in nature. They are biodegradable and allow quick drug uptake to the brain. However, they also have some disadvantages, including their inability to solubilize substances with a high melting point, many times requiring large quantities of surfactants to stabilize the system (which may compromise their safe use), and a high production price [48,51,52].

On the other hand, nanoparticles are solid substances with a size that can vary between 10 and 1000 nm that allow to dissolve, encapsulate, absorb, or attach hydrophilic or lipophilic drugs (depending on the characteristics and composition of the nanoparticle). These systems have several advantages, such as: making it possible to avoid drug degradation, consequently increasing its concentration at the action site; having a reduced particle size, which results in higher surface area, allowing to overcome biological and physiological barriers; allowing drug targeting according to the nanoparticle’s surface functionalization, directing it to the site of action. In intranasal administration, a particle size between 10 and 300 nm is advantageous since nanoparticles that are this size can be transported directly by the olfactory nerve to the brain. There are two major groups of nanoparticles: polymeric and lipid [33,46,48].

Polymeric nanoparticles can be divided into two groups, depending on the preparation method that is used and on the characteristics of the resulting system: nanocapsules, in which there is a reservoir system, which means the drug is confined to a cavity surrounded by a single polymer membrane that is present on the surface of the nanoparticle, coating it; or nanospheres, which are homogeneous matrix systems, in which the drug is uniformly dispersed or dissolved in. By modulating the polymer, it is possible to control the release of the drug to reach the desired therapeutic concentrations at the action site for the required period of time [33,48].

Lipid nanoparticles are subdivided into solid lipid nanoparticles (SLNs) and nanostructured lipid carriers (NLCs). SLNs are constituted by a solid lipid matrix, that is, by lipids that remain solid at room temperature and body temperature. The lipophilic drug is dissolved or dispersed in this matrix, which is surrounded by a surfactant layer in an aqueous dispersion that contributes to the stability of the nanoparticle. When compared to polymeric nanoparticles, lipid nanoparticles are considered more biocompatible and biodegradable, have low toxicity, are easy to produce on a large scale, have better physical stability, and good control of drug release. SLNs, however, have some disadvantages, namely the structural reorganization they undergo over time (such as recrystallization), because they only have one type of lipid, which creates a tighter internal structure with less space to incorporate drug molecules. In order to overcome this problem, the NLCs emerged, which have a matrix formed by a solid lipid and a liquid lipid. The addition of liquid lipids prevents crystallization and allows structural disorganization in the lipid matrix, creating larger spaces to incorporate the drug molecules in. Yet, despite the advantages of the NLC, SLN remains an effective system. Since the mucosa has a lipid nature, lipid nanoparticles can be transported by passive diffusion to the site of action [42,48,50].

Thus, nanometric drug transport systems, and specifically nanoparticles, present advantages for intranasal drug administration since they allow to protect the drug from mucociliary clearance and the enzymatic degradation that occurs in the nasal cavity, improve its absorption, and allow to modulate the time and quantity of drug available at the site of action, favoring transport through the biological membranes [30,53]. A summary of the ideal nanoparticle formulation characteristics for intranasal administration is depicted in Figure 3.

Taking into account the increased incidence of mental illness, and in particular depression and anxiety, an increasing need for safer, faster, and more effective treatments has been identified. Analyzing the advantages of intranasal administration together with the advantages of nanoparticles, the development of new formulations using nanoparticles for intranasal administration seems to have great potential as an alternative to existing treatments. The following sections will summarize and do a critical analysis of the already existing studies regarding the development and evaluation of nanoparticles for intranasal administration of antidepressant and anxiolytic drugs that may contribute to a more effective future treatment of these pathologies.

## 2. Nanoparticles as Efficient Strategies to Deliver Antidepressant and Anxiolytic Drugs to the Brain

### 2.1. Icariin

Icariin is a component of the aerial parts of the plant *Epimedium brevicornum* Maxim (Berberidaceae), which presents antidepressant-like effects. After oral administration, the reached plasma concentrations are very low, with the drug being weakly absorbed by this route, limiting its use in the treatment of depression.

A study conducted by Xu et al. [54] aimed to develop a nano thermoresponsive hydrogel (nanogel) to allow the encapsulation of icariin for intranasal administration in order to increase the amount of drug that reaches the brain, and to verify its antidepressant-like activity. Nanogel was obtained using alginate, an ionic polysaccharide with hydrophilic characteristics that allows the increase in the weak solubility of icariin in water, and divalent cations, which, when in contact with alginate, will enable the formation of reversible hydrogels (Figure 4A). Additionally, poloxamer 407 and poloxamer 188 were added to the developed hydrogel, due to their thermosensitive characteristics, with the objective of improving the bioavailability of icariin and stabilizing the nanosystem. Therefore, polymeric alginate nanoparticles were formulated, being placed in a matrix of poloxamers right after. Due to the presence of alginate, a mucoadhesive polymer, the developed semisolid formulation also had the ability to adhere to the nasal mucosa. The particle size of the icariin nanogel (73.80 nm) was smaller than the particle size of the drug-free vehicle (87.28 nm), which may be due to the fact that icariin is encapsulated in the nanogel through hydrogen bonds, with superior intermolecular forces. The polydispersity index (PDI, a particle size homogeneity index ranging from 0 to 1) and zeta potential (particle surface charge) of the optimized formulation were 0.15 and −19.2 mV, respectively, which means that the formulated nanogel presented homogeneous particle size and was potentially stable due to electrostatic repulsion between the charged nanoparticles.

In vitro drug release studies showed that the nanogel had a controlled release, allowing the drug to be fully released after 36 h. The antidepressant-like effects of the formulated nanogel were studied by conducting in vivo pharmacodynamic studies in mice, such as the forced swim test (Figure 4B), tail suspension test (Figure 4C), and open field test (Figure 4D). The results showed that the formulated nanogel allowed to reduce the immobility time of the mice after only one intranasal administration when compared with the oral administration of a drug solution that required administration for seven consecutive days. The results support that the antidepressant effect is the fastest with the intranasal administration of the nanogel. In vivo studies were also conducted in rats, using the CUMS (chronic unpredictable mild stress) model, which, when applied, leads to a reduction in body weight and decreases the preference for sucrose in depressed animals. The treatment with the formulated intranasal nanogel allowed to reverse these effects, overall increasing the rat’s body weight (Figure 4E) and restoring their sucrose intake more significantly than any other treatment group (Figure 4F).

This study allowed us to conclude that the formulated icariin nanogel, when administered intranasally, presents relevant potential as a possible alternative for the treatment of depression.

### 2.2. Albiflorin

Albiflorin is the main component of the root of *Radix Paeoniae Alba* (Ranunculaceae), presenting antioxidant, anti-inflammatory, and neuroprotective properties and antidepressant-like effects. Nevertheless, when administered orally, it has low bioavailability, not reaching effective brain concentrations to evidence a significant antidepressant-like effect.

To tackle these issues, a study conducted by Xu et al. [55] aimed to develop an intranasal albiflorin nanogel. Once again, the interaction of alginate with divalent cations was used for the formulation of the nanoparticles, with the subsequent addition of poloxamer 407 and poloxamer 188 to form the nanogel and in order to stabilize the system and allow a controlled release of the drug. The particle size of the nanogel was small (45.6 nm), as was the PDI (0.20), making it quite a homogeneous system. The obtained zeta potential was negative (−19.8 mV) due to the presence of alginate, a polyanion, in the composition of the nanoparticle.

In-vitro drug release studies showed a controlled release of the drug from the nanogel, being extended in time for about 12 h. Moreover, in vivo pharmacokinetic studies in rats, the drug was not only rapidly quantified in the brain, but also allowed the maintenance of prolonged concentrations over time, which indicated a rapid transport and a potential prolonged therapeutic effect. In-vivo pharmacodynamic studies were also conducted, in mice, with the purpose of evaluating the antidepressant activity of the formulation. The tail suspension test demonstrated that the intranasal administration of the nanogel significantly reduced the immobility time in mice when compared with intragastric or intravenous administration of fluoxetine (antidepressant drug of known efficacy) or albiflorin solutions. The CUMS model was applied in rats, causing a decrease in body weight as well as in sucrose intake, but these effects were once again reversed by intranasal administration of the albiflorin nanogel.

In conclusion, the formulated nanogel was able to encapsulate albiflorin and, when administered intranasally, provide a continuous and controlled release of the drug that made it possible to evidence its antidepressant-like effect, presenting potential as an alternative for the treatment of depression.

### 2.3. Fluoxetine

Fluoxetine is a selective serotonin reuptake inhibitor. The main problem associated with this drug is that it is prone to drug-drug interactions when administered orally since it is a CYP2D6 inhibitor and a substrate and inhibitor of the P-glycoprotein efflux transporter.

To tackle these issues, Vitorino et al. [56] aimed to develop a formulation based on lipid nanoparticles for intranasal administration of fluoxetine in order to find the optimal conditions to combine a rapid onset of action with a prolonged therapeutic effect. Initially, the components were screened, taking into account the relative solubility of fluoxetine in them. As a solid lipid, glyceryl palmitostearate (Precirol™ ATO 5), a biocompatible glyceride, was used. The selected liquid lipid was propylene glycol monocaprylate (type I) (Lauroglycol™ 90). Polysorbate 80 (Tween^®^ 80) was used as a surfactant due to its good emulsifying capacity, biocompatibility, and ability to stabilize the nanosystem. In the optimization stage of the right component proportions (liquid lipid: solid lipid ratio and amount of surfactant), the main conclusions were: that formulations with higher surfactant concentration decreased permeability and extended drug release; that an increase in the amount of liquid lipid resulted in increased release and decreased permeability; that the increase in the concentration of liquid lipid and surfactant allowed to have smaller particles and a more homogeneous size; and that the formulation with the highest concentration of solid lipid and the lowest amount of surfactant achieved the best permeability. Thus, the optimized formulation had similar percentages of solid lipid and liquid lipid and low surfactant concentration, presenting a particle size of 154 nm, PDI of 0.514, and zeta potential of +19.7 mV. Although the particle size was small, and the zeta potential had a moderately high absolute value (which could contribute to particle stability), the PDI value was high, making this a reasonably heterogeneous formulation. The encapsulation efficiency (EE%) (amount of drug that is possible to incorporate in the lipid matrix) and drug loading (percentage of encapsulated drug divided by the total mass of the lipid matrix) were also determined, being around 74% and 13%, respectively.

In-vivo pharmacodynamic studies were conducted in mice, namely the forced swim test, applied one hour after the administration of the formulations. This allowed us to verify that both the intranasal nanoparticles and an oral fluoxetine solution (positive control) led to an increase in the mobility time when compared to the group of untreated animals. However, the immobility time was shorter after oral administration of the fluoxetine solution. These results may be related to the time elapsed between the administration of the drug and the performance of the test, since, in general, the drugs administered intranasally can take about 15 min to reach the brain

Based on the obtained results, since no comparative superiority was demonstrated in this study it is possible to conclude that the formulation of lipid nanoparticles administered intranasally presents potential as an alternative to oral administration when the oral route is not accessible.

### 2.4. Agomelatine

Agomelatine is a synthetic compound derived from melatonin used in the treatment of depression. It is an agonist of the melatonin MT1 and MT2 receptors and an antagonist of 5-HT2C serotonin receptors. This drug undergoes a substantial hepatic first-pass metabolism when administered orally, leading to an extremely low bioavailability. The study conducted by Jani et al. [57] aimed to solve these issues by analyzing the efficacy of the intranasal administration of agomelatine polymeric nanoparticles for the treatment of depression.

The formulated polymeric nanoparticles were obtained using the polymer PLGA, due to its low toxicity and poloxamer 407 as a surfactant to stabilize the nanoparticles. The optimization of the formulation considered the drug: polymer ratio and surfactant concentrations since they affect the particle size and the EE%. A statistical study allowed us to verify that by increasing the amount of PLGA the EE% also increases; however, after a certain amount of PLGA the EE% begins to decrease. The same was verified for the relation between surfactant concentration and EE%. Regarding particle size, it was concluded that it increases with the increase in the amount of PLGA, and decreases with the increase in the amount of surfactant. Thus, the optimized formulation aimed at the highest EE% value (98.3%) and the lowest particle size (116.06 nm). The zeta potential of the optimized formulation was −22.7 mV, and the PDI was less than 0.3, indicating that the formulated system was stable and the particle size distribution was homogeneous. Drug loading (49.2%) was also calculated.

In vitro drug release studies showed that the release of drug from the nanoparticles was more prolonged than the free drug, indicating an extended drug release, which can be seen as an advantage. Ex vivo permeation studies were performed using goat nasal mucosa, and the results allowed us to conclude that the permeability of the formulation was also superior to the permeability of a free drug suspension.

The forced swim test (in vivo pharmacodynamic study) was performed in rats and allowed to verify that, although the two analyzed formulations (PLGA nanoparticles and simple drug suspension) were able to reduce immobility time, the PLGA nanoparticles demonstrated the most significant reduction when compared to a group of untreated animals.

Thus, given the results of in vitro drug release, ex vivo drug permeation, and in-vivo pharmacodynamic studies, it is possible to conclude that the formulated nanoparticles have the potential for the treatment of depression when administered intranasally.

### 2.5. Venlafaxine

Venlafaxine is a serotonin and norepinephrine reuptake inhibitor. The oral administration of this drug is associated with several adverse effects, such as headaches and dizziness, among others, with low bioavailability and a short half-life time. Additionally, frequent administrations are necessary to produce the desired effect [37,38,39].

A study done by Haque et al. [58] aimed to tackle these issues by verifying the potential of chitosan nanoparticles to improve the transport of venlafaxine to the brain, using intranasal administration. The formulated polymeric nanoparticles were composed of chitosan, a polycation, and sodium tripolyphosphate (TPP), a polyanion. The optimized formulation considered a drug: polymer ratio of 1:1, and also took into account the proportion between chitosan and TPP concentrations since it affects the formulation of nanoparticles. The obtained nanoparticles presented a particle size of 167 nm, a PDI of 0.367, a zeta potential of +23.83 mV, and a pH of 5.12, thus concluding that the formulated system had a surface charge that increased its stability, with particle size and pH adequate for intranasal administration, although not being very homogeneous. It also presented an EE% of 79.3% and drug loading of 32.3%. In vitro drug release studies allowed us to conclude that the formulated nanoparticles presented a biphasic release profile, for which in the first 2 h there was a rapid release, followed by an extended release over 24 h. Drug permeation was analyzed in ex-vivo studies using pig nasal mucosa, and the use of chitosan nanoparticles allowed us to increase the permeability of venlafaxine 3-fold when compared with a venlafaxine solution. In vivo pharmacodynamic studies in rats (forced swim test) allowed us to conclude that the intranasal administration of the developed chitosan nanoparticles, when compared with the control group (saline solution), increased swimming time and locomotor activity and reduced immobility time. Pharmacokinetic parameters were also evaluated, such as the maximum concentration of venlafaxine in the brain and brain AUC, and the results were better for the intranasal nanoparticles (when compared with an intravenous or intranasal venlafaxine solution), and the intranasal route in general (when compared with the intravenous drug solution). These results could be attributed to the action of chitosan in paracellular transport enhancement and alternate opening of the tight junctions, and also its contribution to the reduction of mucociliary clearance. Brain targeting ratios, such as direct transport percentage (DTP%) and drug targeting efficiency (DTE%), were also calculated, presenting high values for the formulated nanoparticles (80.34% and 508.59%, respectively), supporting their capacity for brain targeting. Thus, the formulated nanoparticles, administered intranasally, were successful in increasing the transport of the drug from the nasal cavity to the brain, thus demonstrating potential as an alternative to the oral administration of venlafaxine.

Another study made by Haque et al. [59] aimed to analyze the potential of the intranasal administration of venlafaxine alginate and chitosan nanoparticles for the treatment of depression. To obtain the formulation, firstly, alginate was added to calcium chloride, which allowed the occurrence of inotropic pre-gelation, obtaining a pre-gelified nanonucleus, within which the drug was encapsulated. Then chitosan (positively charged) was added, which interacted with the negative charge of alginate, originating the nanogel. The optimization of the formulation took into account the concentrations of alginate, calcium chloride, and chitosan, and the drug: alginate ratio was set at 0.75:1. The pH was maintained between 5.7 and 6.1, being carefully optimized, since a higher pH results in chitosan precipitation, making it less available for the formation of the nanoparticles. Additionally, the addition of an alginate solution at neutral pH results in the non-protonation of the chitosan amine groups, stopping its ionic interaction with alginate. Particle size (173.7 nm), zeta potential (+37.4 mV), and PDI (0.391) allowed to confirm the reduced size and potential stability of the nanosystem, albeit being slightly heterogeneous. Drug loading (26.74%) and EE% (85.6%) were also determined. In-vitro drug release studies showed that the formulated nanoparticles had a biphasic release profile, with an initial faster release, followed by an extended release. Ex vivo permeation studies (pig nasal mucosa) showed that the formulated nanoparticles had an improved permeability when compared to a drug solution. In vivo pharmacodynamic studies (in rats), namely the forced swimming and locomotor activity tests, showed that, when compared with the control group (intranasal administration of a saline solution), the intranasal nanoparticles allowed to increase swimming time and locomotor activity and reduce immobility time. The pharmacokinetic parameters were also evaluated, and the brain concentrations and AUC of venlafaxine were significantly higher after intranasal administration of the nanoparticles when compared with intravenous administration and intranasal administration of a venlafaxine solution. DTE% (425.77%) and DTP% (76.52%) allowed us to conclude that intranasal administration of formulated nanoparticles also allowed a better brain targeting of the drug when compared with all other groups. Therefore, the study revealed that the formulated nanoparticles, administered intranasally, allowed us to achieve prolonged venlafaxine concentrations in the brain, probably due to an improvement in paracellular transport through the modulation of tight junctions between cells, presenting potential in the treatment of depression.

A third study, conducted by Cayero-Otero et al. [60], also aimed to find an intranasal nanosystem that would enable the controlled release of venlafaxine, maintaining therapeutic levels for a prolonged period of time. PLGA polymeric nanoparticles were developed, having two distinct ligands: transferrin (Tf) and a specific peptide against Tf receptor (TfRp). Ligands allow for the modification of the surface of nanoparticles, improving absorption and increasing permeability due to a better interaction between the cells and the nanosystems, allowing a targeted delivery. The non-functionalized nanoparticles (without ligand) originated with a particle size of 206.3 nm, PDI of 0.190, and zeta potential of −26.5 mV. The nanoparticles with Tf and the nanoparticles with TfRp originated a particle size of 218.6 nm and 216.3 nm, PDI of 0.078 and 0.067, and zeta potential of −19.5 mV and −19.6 mV, respectively. Hence, it is evident that although the ligands slightly increase the particle size, they also reduce the PDI, leading to more stable and homogeneous nanoparticles. EE% ranged from 48 to 50%, and drug loading from 10 to 12% for all formulations. The three formulations demonstrated a biphasic drug release profile. Initially, there was a fast release, and after that, there was a prolonged release for 10 days. It was possible to observe that the non-functionalized nanoparticles presented an overall higher cumulative drug release when compared to the functionalized nanoparticles. Hence, the ligands led to a decrease in drug release, probably due to the forming of an additional barrier on the surface of the nanoparticle. In-vivo pharmacokinetic studies, it was observed that the mice brain drug concentration 30 min after intranasal administration was much higher for the non-functionalized nanoparticles. This was justified by the authors as being due to the type of transport adopted by the different nanoparticles since the non-functionalized nanoparticles were preferably transported by the olfactory route, through facilitated transport or extracellular transport, taking only a few minutes to reach the brain; and the functionalized nanoparticles were preferably transported by endocytosis, which is a slower type of transport. Hence, compared to non-functionalized nanoparticles, functionalized nanoparticles take longer to reach the therapeutic action site, which can be a disadvantage if a rapid onset of action is desired. However, they have a more controlled release and may allow more prolonged therapeutic action. Hence, it was concluded that both the non-functionalized and functionalized developed nanoparticles had the potential to provide a controlled release of venlafaxine when administered intranasally and may be candidates as an alternative to the oral administration of this drug.

Comparing the three aforementioned studies, it is possible to observe that all of them formulated polymeric nanoparticles for venlafaxine encapsulation. The first two studies [58,59] had a particle size lower than 200 nm, but a PDI value higher than 0.3, which indicates that the particle size is small but not very homogeneous (considering an optimal value of < 0.3). The Zeta potential of both studies was positive. The third study [60] had an additional evaluation of the effect of surface ligands, and both functionalized and non-functionalized nanoparticles had a particle size higher than 200 nm, PDI less than 0.3 (which is indicative of homogeneity), and a negative zeta potential value. In vitro release studies of all the developed nanoparticles indicated a biphasic drug release, with an initial faster onset, followed by a prolonged release. In vivo pharmacodynamic studies also showed favorable results, supporting that the formulations allowed to potentiate the antidepressant action of venlafaxine. All the formulations developed in the three mentioned studies showed potential in the treatment of depression.

### 2.6. Desvenlafaxine

Desvenlafaxine is an active metabolite of venlafaxine, a second-generation serotonin and norepinephrine reuptake inhibitor. It has an oral bioavailability of about 80% and a half-life time of about 11 h; however, it is associated with several adverse symptoms such as increased blood pressure, nausea, and headaches, among others. In order to tackle these issues, a study conducted by Tong et al. [61] aimed to develop biodegradable and biocompatible polymeric nanoparticles for intranasal administration of desvenlafaxine for depression treatment.

The formulated polymeric nanoparticles were composed of: the polymer PLGA; chitosan, which was used due to its mucoadhesive properties; and polyvinyl alcohol (PVA), which in addition to chitosan, has the ability to stabilize the formulation. Particle size, PDI, zeta potential, EE%, and drug loading were, respectively, 172.5 nm, 0.254, +35.63 mV, 98.3%, and 49.15%. In-vitro drug release studies showed an initially fast drug release, probably of a drug present at the surface of the nanoparticles, followed by an extended release during 24 h, which probably corresponded to the fraction of the drug that was inside the nanoparticle, in its nucleus, by hydration and swelling of the nanoparticle matrix. A pH of 7.4 was considered for optimized nanoparticles. Also, there were no differences in drug release at pH 7.4 (physiological pH) when compared to pH 6.0 (pH within the range of nasal mucosa values).

In vivo pharmacokinetic studies in rats (Figure 5A), with intranasal and intravenous nanoparticle administration, and intranasal administration of a desvenlafaxine solution, showed that desvenlafaxine brain concentrations, half-life time (*t*_1/2_) and AUC were higher when the administration was intranasal, regardless of the formulation that was used. Furthermore, these parameters were superior for the intranasal desvenlafaxine nanoparticles when compared to the drug solution. These results support intranasal administration for better pharmacokinetic parameters in the brain, as well as the use of nanoparticles for the incorporation of the drug. Additionally, the values of DTE% and DTP% were calculated and were higher for the formulated nanoparticles administered intranasally (544.23% and 81.62%, respectively), which indicates better efficiency in brain targeting when compared to the other comparative groups.

In-vivo pharmacodynamic studies (Figure 5B), such as the reserpine reversal test and the forced swim test, were conducted, administering the optimized nanoparticles intranasally or a desvenlafaxine solution intranasally or orally. The results were compared with the ones from a control group (intranasal administration of a saline solution). It was observed that only the intranasally administered formulations had a significant effect on the ability to reduce reserpine-induced immobility. Furthermore, the Forced Swim Test allowed to conclude that the intranasal nanoparticles had a more significantly reduced immobility and increased swimming, climbing, and locomotion time than all other groups.

Monoamine levels in the brain, and their association with antidepressant activity in different groups of rats subjected to the forced swim test, were also studied (Figure 5C). The results showed that the intranasal nanoparticles significantly improved serotonin and norepinephrine levels in the brain, supporting the theory that the levels of these two neurotransmitters decrease in stressful situations, particularly in depression.

In conclusion, the formulated nanoparticles, administered intranasally, allowed us to improve the pharmacokinetic and pharmacodynamic profile of desvenlafaxine, helping with the transport of the drug to the site of action in the brain, and showing promise as an alternative for depression treatment.

### 2.7. Selegiline

Selegiline is a MAOi, with a dose-dependent therapeutic action and also presents neuroprotective effects. When administered orally, this drug undergoes extensive hepatic metabolism, having reduced bioavailability and also several associated adverse effects. A study conducted by Singh et al. [62] aimed to increase the efficacy of selegiline (hydrochloride form) through intranasal administration of thiolated chitosan nanoparticles in order to obtain an extended and controlled release of the encapsulated drug.

The formulated nanoparticles contained thiolated chitosan, which is obtained by coupling the primary amine groups of chitosan with thiol groups. This structural change improves the mucoadhesiveness and permeability enhancement capability of this polymer. TPP was also used as a polyanion that interacts with chitosan through electrostatic forces, causing chitosan to precipitate, forming spherical particles. The optimization of the formulation considered the thiolated chitosan: TPP ratio since it directly affects the particle size and its distribution. The optimized ratio was 5:1. The pH was also considered, being ideal between 5.0 and 5.5 (values in the range of the nasal mucosa). Also, with a pH above 5.5, there was the formation of unwanted white aggregates that disappeared when a reduction of pH was verified.

In vitro drug release was analyzed and compared between thiolated chitosan nanoparticles and unmodified chitosan nanoparticles. It was concluded that the unmodified nanoparticles showed a faster release profile in the first 2 h, due to the presence of a drug on the surface of the nanoparticles that is released faster when compared with thiolated chitosan nanoparticles, for which there was a formation of an in situ gel. This means that there is a high-density gel network that restricts the penetration of water at the site where the diffusion of the drug occurs, delaying the release. Nevertheless, after about 3 h, it was found that the release of the drug from thiolated chitosan nanoparticles was constant and extended, becoming higher than the release of unmodified chitosan after 13 h.

In-vivo pharmacodynamic studies in rats (stress-induced immobility, sucrose preference test, and locomotor activity), the intranasal administration of the thiolated chitosan nanoparticles was compared to the also intranasal administration of unmodified chitosan nanoparticles or a drug solution. The drug solution only showed significant effects on increased locomotor activity. Both nanoparticle types allowed reduced immobility time, restored sucrose intake, and increased locomotor activity, with better results being obtained after the administration of thiolated chitosan nanoparticles.

Additionally, another evaluated parameter was the formation of free radicals, namely nitrite, since oxidative stress has been associated with mood disorders. Its concentration was more significantly reduced when thiolated chitosan nanoparticles were administered, compared to the administration of the drug solution or the unmodified chitosan nanoparticles.

In conclusion, the formation of an in situ gel allows the improvement of the permeability and transport of the drug from the nasal cavity to the action site in the brain, and there was a significant advantage when using thiolated chitosan when compared to unmodified chitosan.

### 2.8. Tramadol

Tramadol is a synthetic opioid drug with central analgesic action. Aside from its affinity for opioid receptors, it is also able to inhibit the reuptake of the neurotransmitters serotonin and norepinephrine, thus presenting an antidepressant effect. Nevertheless, it is quite prone to metabolism; hence, the incorporation of tramadol in nanoparticles could allow protecting the drug from its labile nature. A study conducted by Kaur et al. [63] aimed to study the efficacy of the transport of tramadol to the brain using the intranasal administration of nanoparticles incorporated in an in situ gel for the treatment of depression.

The polymeric nanoparticles were composed of chitosan and TPP, and incorporated in a thermosensitive in situ gel made of poloxamer 407 and hydroxypropylmethylcellulose (HPMC) K15M. Benzalkonium chloride was also used as a cationic surfactant and as a stabilizer of the system. Chitosan and TPP were used in an optimized ratio of 1:1, and the optimized nanoparticles presented a particle size of 152.0 nm, PDI of 0.143, the zeta potential of +31 mV, and EE% of 85%. In-vitro drug release studies demonstrated a biphasic release of the drug, and the nanoparticles incorporated into the in situ gel presented a longer drug release when compared to a saline solution.

The forced swim test (in vivo pharmacodynamic study) was performed in rats, and it was possible to observe that the intranasal administration of the nanoparticles incorporated in the in situ gel allowed to reduce the immobility time and increase the locomotor activity. Body weight was also assessed before and after the treatment, and a significant improvement was observed after the administration of the nanoparticles when compared to the control group (no treatment). Furthermore, the sucrose preference test allowed us to verify that the intranasal nanoparticles in situ gel increased the intake of sucrose. Nitrite concentration was also evaluated, being significantly reduced after the treatment with the nanoparticles. It should also be noted that, in all these studies, the intranasal administration of a tramadol saline solution obtained unsatisfactory results.

Hence, in general, the developed tramadol nanoparticles in situ gel was able to ensure a controlled release of the drug, improving transport to the brain, and proving itself to be a possible alternative to oral administration of the drug.

### 2.9. Buspirone

Buspirone is an anxiolytic drug used in the treatment of generalized anxiety disorder. It acts as an agonist of pre-synaptic serotoninergic receptors in the hippocampal region of the brain, and as a partial agonist of post-synaptic serotoninergic receptors, in the raphe nucleus. It undergoes an extensive first-pass hepatic metabolism when administered orally, presenting reduced bioavailability. The study by Bari et al. [64] aimed to tackle this problem by assessing the efficacy of the brain transport of polymeric nanoparticles with the encapsulated drug, after intranasal administration, for the treatment of anxiety.

The formulated nanoparticles were either composed of thiolated chitosan or plain chitosan and also a mixture of TPP and alginate, which was used as a cross-linker. The optimization of the formulation considered the relationship between chitosan and cross-linker concentrations since it affects particle size and PDI values. The optimized chitosan and thiolated chitosan nanoparticles presented, respectively, a particle size of 195.7 nm and 208.3 nm and a PDI of 0.367 and 0.253. The decrease in PDI value indicates greater homogeneity of the system due to the presence of thiolated chitosan. It was also verified, by scanning electron microscopy (Figure 6A), that both formulations presented pores on the surface, which suggests the possibility of drug release by these same pores.

In vitro drug release was compared between the two formulations, and it was observed that the thiolation of chitosan did not affect it. Both formulations showed a biphasic release, with an initial fast release, followed by an extended release for 24 h, with more than 90% of the drug being released after that time. Nevertheless, ex vivo permeation studies (Figure 6B) showed that thiolated chitosan nanoparticles had better permeability when compared to unmodified chitosan nanoparticles. These results can be explained by the increased mucoadhesion since a strong ionic bond is formed between the thiolated chitosan amine group and the cell membrane sites, which results in the opening of the tight junctions.

Pharmacokinetic parameters were determined in vivo studies (Figure 6C) after intranasal administration of the thiolated chitosan nanoparticles or a buspirone solution or intravenous administration of the same drug solution. The brain concentration and AUC for the intranasal nanoparticles were higher than for all other groups at all points in time. It should also be noted that the concentration of buspirone in the brain that was obtained 30 min after administration was about three times higher for the intranasal nanoparticles, which indicates that this route of administration allows a faster reaching of the CNS bypassing the blood-brain barrier. DTP% (95.97%) and DTE% (78.94%) values were also higher after intranasal administration of the formulated nanoparticles (compared with the drug solution, also administered intranasally), further showing their superiority.

Therefore, it can be concluded that the developed polymeric nanoparticles are a possible noninvasive alternative to buspirone administration for the treatment of generalized anxiety disorder.

### 2.10. Riluzole

The mechanism of action of riluzole is not completely defined. It is assumed that it acts as an inhibitor of glutamate release and that it is associated with decreased oxidative stress, presenting the potential for the treatment of anxiety. This substance is subjected to an extensive first-pass effect metabolism when administered orally, which results in the reduction of its bioavailability. The study conducted by Nabi et al. [65] aimed to find a solution for this issue by developing polymeric nanoparticles for intranasal administration containing encapsulated riluzole in order to increase the concentration of the drug in the brain and to enhance its action in reducing oxidative stress.

Chitosan was used as a polymer in the formulation of the nanoparticles due to its mucoadhesive properties, and TPP was used as a polyanion. For comparative purposes, chitosan nanoparticles were formulated with and without Tf. Tf was added as a ligand in order to facilitate the transcytosis of the nanoparticles through the blood-brain barrier (for a fraction of the drug that will undergo indirect transport). The optimized formulations considered the drug:polymer: Tf ratio. The particle size and PDI for chitosan nanoparticles were 173.6 nm and 0.264, and for chitosan nanoparticles with Tf were 207.0 nm and 0.406, respectively. It was possible to observe an increase in particle size and PDI with the addition of Tf; therefore, nanoparticles without the ligand presented a more homogeneous system. The optimal pH value chosen for the formation of nanoparticles with Tf was 5.0, a value that is considered to be compatible with the nasal mucosa. In vitro drug release was shown to be initially fast, followed by an extended release. The cumulative release after 24 h was higher for the nanoparticles in which the ligand was used, followed by the nanoparticles without the ligand, and then a drug suspension. The same conclusions were reached in ex vivo permeation studies, with the nanoparticles with the ligand having higher permeation, followed by the nanoparticles without ligand, and with drug suspension permeating the least.

Intravenous or intranasal administration of a riluzole suspension and intranasal administration of the nanoparticles with Tf was done in rats to assess their pharmacokinetics. It was possible to observe higher brain drug concentrations for intranasal administration, in general, with it being more significant for the nanoparticles. The results also supported the hypothesis that the nanoparticles administered intranasally have an extended drug release, as seen in prolonged brain drug levels. DTE% (1138.46%) and DTP% (91.21%) values showed better results for the nanoparticles that contained the ligand, so it was concluded that the ligand allowed improved efficiency of brain drug transport. Pharmacodynamic studies conducted in rats (elevated plus maze model and Morris water maze test), showed that riluzole has substantial anxiolytic activity and that nanoparticles with and without ligands were able to improve learning capacity and memory in the animals subjected to the study. It was also possible to conclude that the treatment with both formulated nanoparticles (with and without ligand) was able to significantly reduce oxidative stress (although for this test, no significant difference was observed between the two types of nanoparticles).

Therefore, the formulated nanoparticles demonstrated potential for the treatment of anxiety through intranasal administration, with the use of Tf as a ligand being an additional advantage throughout the study, improving brain drug targeting.

## 3. Final Remarks

The current treatments for depression and anxiety disorders have several disadvantages that compromise drug bioavailability, with the amount of drug that reaches the intended site of action, the brain, being quite low. This can consequently compromise the therapeutic response. These treatments are also associated with several systemic adverse effects. Therefore, the intranasal route has been studied throughout these past years due to its great potential for improving the current therapies for these pathologies, bypassing their disadvantages. The administration through the intranasal route establishes direct contact between the surrounding environment of the nasal cavity and the CNS, allowing it to overcome the blood-brain barrier and also avoiding the metabolism of the first hepatic passage. This creates a decrease in adverse effects caused by the most prevalent route of administration, the oral route, which brings another major benefit for the patients.

Intranasal drug absorption depends on several anatomical, physiological, physicochemical, and formulation factors, which influence the extent of drug absorption and, consequently, the efficacy of the formulation administered in the nasal cavity. There are several strategies used to improve drug absorption. The analyzed studies show that the incorporation of drugs into nanoparticles is a way to protect the drug and transport it, also allowing a controlled release, which increases and prolongs its concentration in the therapeutical target site. The analyzed studies formulated different types of nanoparticles with different drugs in order to verify the potential of using these formulations, administered intranasally, for the treatment of depression and anxiety.

All studies that reported values of particle size, zeta potential, and PDI, demonstrated that they are, in general, adequate, conferring stability and homogeneity to the system. Moreover, these parameters can highly influence drug absorption and pharmacokinetics since, as already mentioned, nanoparticles with sizes between 10 and 300 nm can undergo direct brain transport (olfactory nerve), and lower PDI values are usually connected with lower variability in drug absorption and distribution. The reported values of EE%, DTE%, and DTP% were also promising, allowing for the achievement of an effective encapsulation of drugs and high brain targeting. The pharmacokinetic and pharmacodynamic studies presented promising results in the application of this type of formulation to produce the desired therapeutic effect. A more detailed comparison between the studies is hampered by the use of different animal models, sample collection times, administered doses, quantification methods, and of course, drugs.

For depression treatment, drugs such as icariin, albiflorin, fluoxetine, agomelatine, desvenlafaxine, venlafaxine, selegiline, and tramadol and, for anxiety, drugs such as buspirone and riluzole were studied. Hence, most studies focused on the treatment of depression. Although both pathologies interfere with the quality of life, depression is considered one of the main causes of disability in the world and can lead to suicide. Therefore, the treatment of this pathology can prevent death, and therefore this could be the reason for a higher focus on this pathology when compared to anxiety disorders.

Polymeric nanoparticles were the chosen nanoparticle type in most studies. This type of nanoparticle allows the control of the release of the drug, and in the majority of studies analyzed, it was reported that a biphasic release occurred: first, a faster release, followed by an extended release, that was maintained for a long period of time. Chitosan and alginate, two mucoadhesive polymers that allow to increase in the residence time of the drug in the nasal cavity and enhance its absorption and arrival at the site of action, were the most used polymers.

Moreover, the functionalization of nanoparticles with ligands, such as Tf or TfRp, as done in the studies conducted by Cayero-Otero et al. [60] and Nabi et al. [65], can allow improving drug absorption and permeability due to facilitated transcytosis of the nanosystems through the blood-brain barrier, which can be quite relevant for a fraction of the drug that will undergo indirect transport. For these functionalized nanoparticles, controlled drug release and permeation was observed, which can allow a more prolonged therapeutic action. In what concerns ligands that could facilitate nose-to-brain drug transport (direct transport), some reports have pointed out that the functionalization of nanoparticles with ligands that have highly expressed receptors in the olfactory region (such as lactoferrin) or glycoproteins (such as lectins) can increase drug transport by facilitated transcytosis and other related mechanisms [66].

Furthermore, the possible clinical application should not be considered until the biocompatibility of the developed nanosystems is assured. Most studies included in this review did not perform any safety studies. Yet, some studies chose their formulation’s composition having in mind the biocompatibility of the used excipients, having searched for safety information priorly (especially from international health authorities, such as EMA and FDA). Although this information could make it more likely that the developed nanoparticles could be potentially safe, one should assess this by actually performing safety studies, for example, in animal nasal mucosa tissues (with the most common models being from pig or sheep) or cells, in order to assess if the developed formulations are actually safe for intranasal administration (no signs of toxicity/irritation/damage). Moreover, in order to increase patient safety, nanotherapeutics should be tailored according to the administration route and disease characteristics (composition compatible with administration site and site of action, adequate particle size, viscosity, pH, etc.). Here it is paramount to make use of adequate analytical tools and thorough prior formulation development planning [67].

Although this review focuses on small molecular weight drugs, intranasally administered biopharmaceuticals have also proven to be effective in the treatment of depression and anxiety disorders. Proteins and peptides such as brain-derived neurotrophic factor, nerve growth factor, neuropeptide Y, insulin, and oxytocin have proven to have potential therapeutic efficacy when administered through the intranasal route, which further shows the potential of intranasal administration [68]. Some of these high molecular weight drugs have also been formulated into nanosystems for intranasal delivery, but, to the best of our knowledge, not for the treatment of depression or anxiety disorders, which leaves scope for further research and development of new macromolecular therapies for these diseases [69,70,71,72,73].

In addition to in-vitro and in-vivo results, recent clinical evidence has surfaced on the effectiveness of the intranasal route for the treatment of major and treatment-resistant depression, with esketamine having proven to be successful, both alone or as adjunct therapy together with oral medication, and with a formulation having even reached the market in 2019, Spravato^®^ [74,75,76,77,78,79]. This further proves the potential of the intranasal route for the treatment of these pathologies and leaves space for the future study of new drugs and formulations to be delivered intranasally.

Despite all the proven potential of the reported nanoparticles for antidepressant and anxiolytic drug delivery, some limitations can hinder the translation of these formulations into the pharmaceutical market. In general, nanoparticles can have scalability issues, meaning that what works at a small laboratory scale might not work on a larger industrial scale, with problems arising in what concerns controlling the formulations desired characteristics (particle size, PDI, zeta potential, EE, etc.) and stability. Furthermore, the complexity of these systems’ composition and production methods could be quite costly and, hence, non-profitable for these companies [67,80]. Moreover, very few nanoparticles for the treatment of depression and/or anxiety have gone as far as clinical trials (and hence none have reached the market), and none using intranasal administration [81]. Yet, nanoformulations for the treatment of several other diseases, such as cancer, multiple sclerosis, or hepatitis, have been successfully developed and approved for commercialization [67,82]. Therefore, further investigation should be done in order to obtain nanosystems with simpler production methods, less expensive components, and scale-up studies could be done to better understand what could happen at an industrial scale so that someday nanoparticles containing antidepressant and/or anxiolytic drugs can reach the market and improve the therapeutic outcomes of these diseases.

Hence, despite the number of studies still being small, in general, the intranasal administration of nanoparticles presents high potential as an alternative to the currently available therapies for the treatment of depression and anxiety. However, the analyzed studies presented only in vitro and in vivo data. Therefore, it is relevant that future research is done in order to compare these animal studies with studies carried out in humans in order to verify the true potential of these formulations when administered intranasally. The focus on research and development of this type of formulation can make treatments increasingly more effective for pathologies that directly interfere with the daily lives of patients, affecting their quality of life.

## 4. Conclusions

In conclusion, the intranasal delivery of nanoparticles for the treatment of depression and anxiety disorders has been proven to be a quite promising approach, allowing to overcome the unavoidable disadvantages of oral administration, reducing the number of administrations necessary for effective treatment, which may increase patient compliance, and also allowing a faster onset of action, with increased and prolonged brain drug concentrations and, consequently, therapeutic effect.

## Figures and Tables

**Figure 1 pharmaceutics-14-02742-f001:**
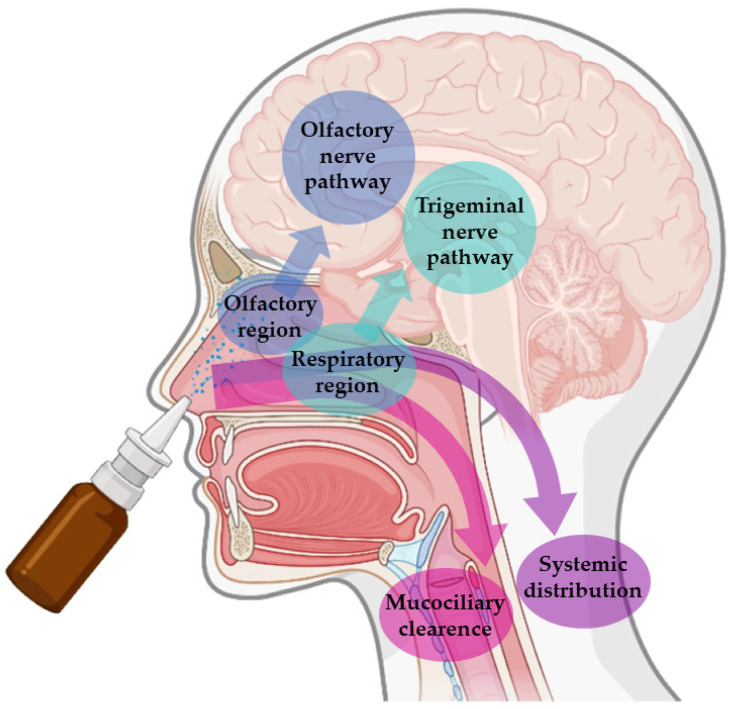
Pathways of brain drug transport after intranasal administration: direct transport (neuronal transport—olfactory nerve and trigeminal nerves) and indirect transport (systemic distribution).

**Figure 2 pharmaceutics-14-02742-f002:**
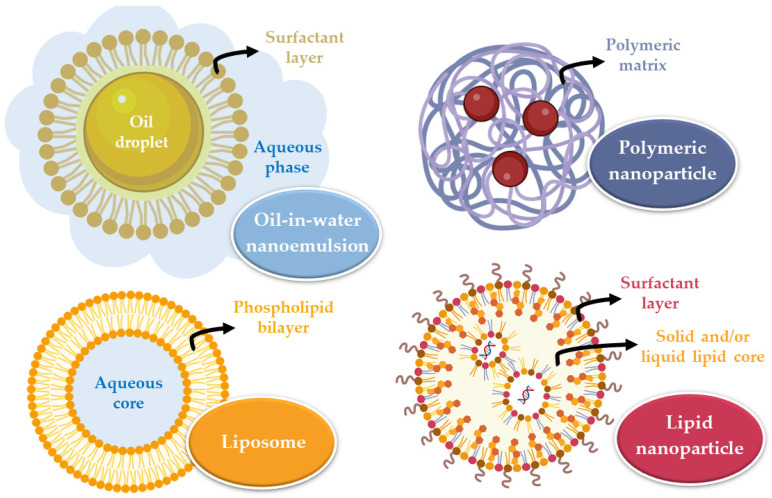
Main types of nanometric systems: liposomes, nanoemulsions, polymeric nanoparticles, and lipid nanoparticles.

**Figure 3 pharmaceutics-14-02742-f003:**
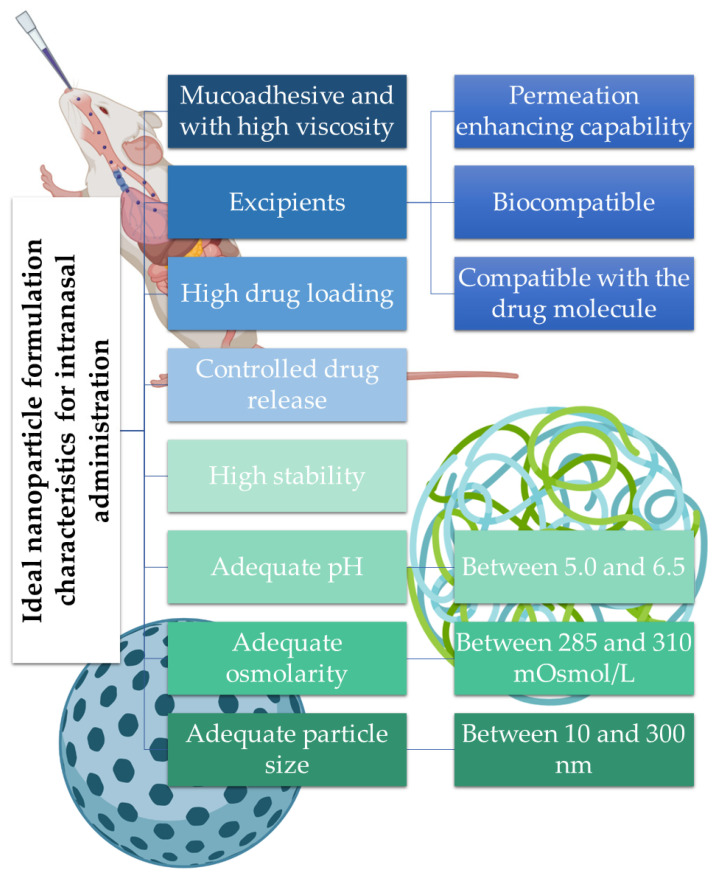
Summary of the ideal nanoparticle formulation characteristics for intranasal administration.

**Figure 4 pharmaceutics-14-02742-f004:**
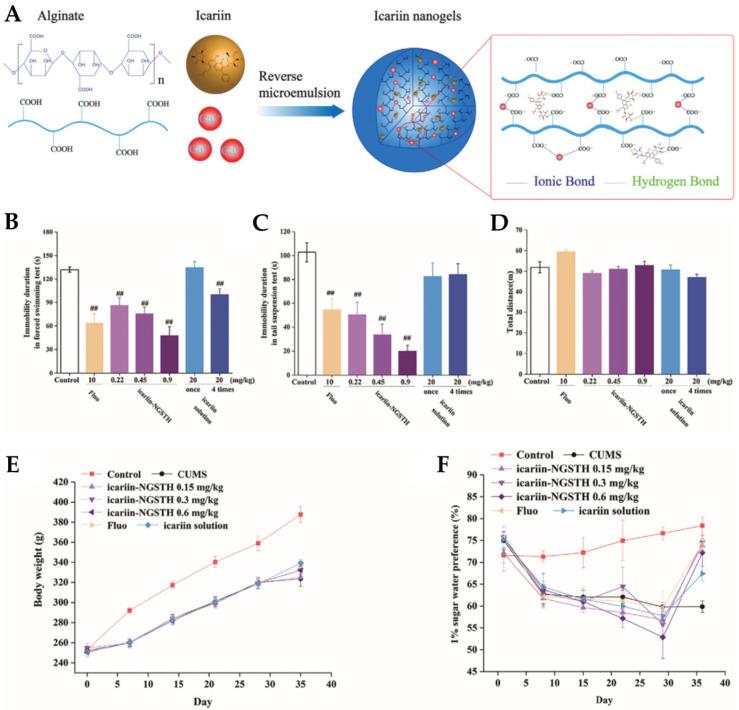
(**A**)—Schematic representation of the making of icariin nanogels; (**B**)—Immobility duration in a forced swimming test, (**C**)—Immobility duration in tail suspension test, and (**D**)—Total distance travelled in open field test, in mice, after intranasal administration of saline buffer (negative control), oral administration of a fluoxetine solution (Fluo, positive control), intranasal administration of the icariin nanogel (NGSTH), or oral administration of an icariin solution; (**E**)—Changes in body weight, and (**F**)—Changes in 1% sucrose preference, in rats, after no administration (CUMS) or intranasal administration of saline buffer (negative controls), oral administration of a fluoxetine solution (Fluo, positive control), intranasal administration of the icariin nanogel (NGSTH), or oral administration of an icariin solution; ## *p* < 0.01, compared with the negative control group; adapted from Xu et al. [54], reproduced with permission from Elsevier [License Number 5426000092048].

**Figure 5 pharmaceutics-14-02742-f005:**
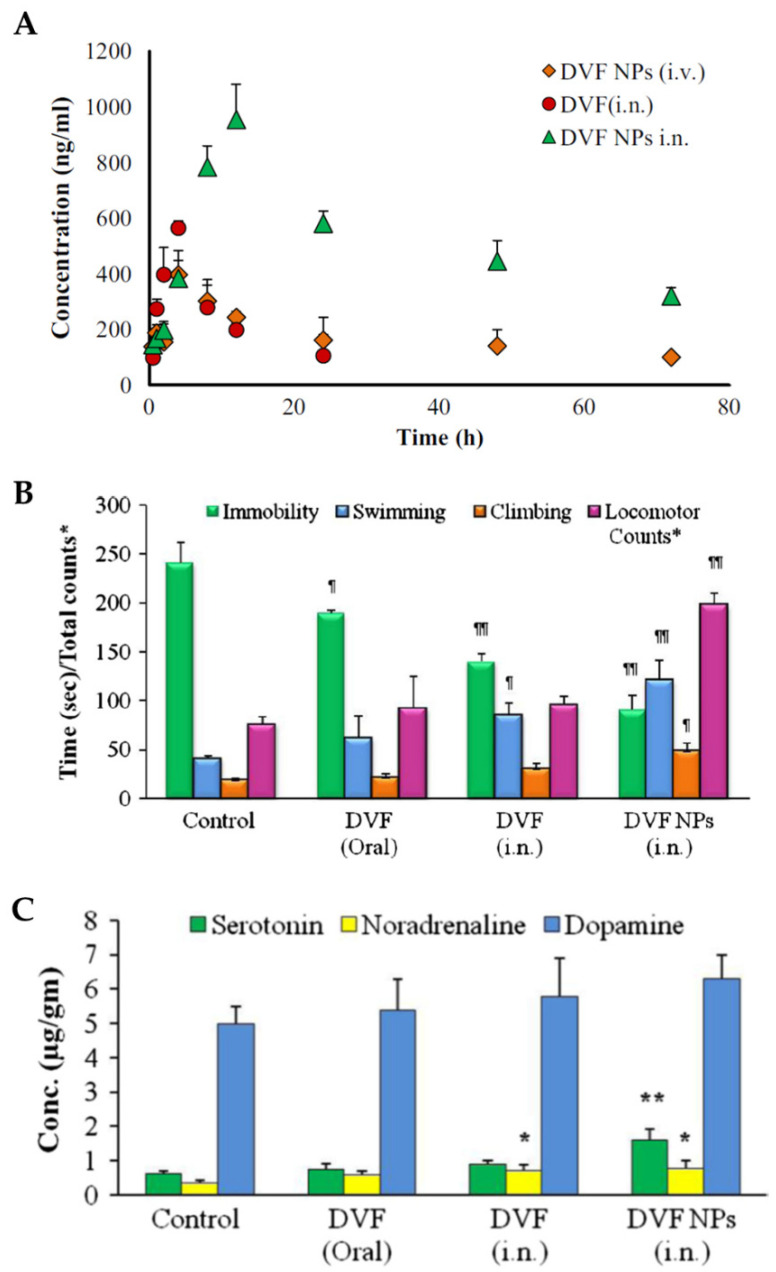
(**A**)—Desvenlafaxine brain concentration after intranasal administration of a drug solution (DVF (i.n.)) or the developed nanoparticles (DVF NPs i.n.), or the intravenous administration of the nanoparticles (DVF NPs (i.v.)); (**B**)—Evaluation of the antidepressant activity of intranasal desvenlafaxine nanoparticles (DVF NPs (i.n.)), intranasal desvenlafaxine solution (DVF (i.n.)) and oral desvenlafaxine solution (DVF (Oral)), in chronically depressed rats; (**C**)—Effect of intranasal desvenlafaxine nanoparticles (DVF NPs (i.n.)), intranasal desvenlafaxine solution (DVF (i.n.)) and oral desvenlafaxine solution (DVF (Oral)) on neurotransmitter levels in rat brain; ¶ and * represent *p* < 0.05, ¶¶ and ** represent *p* < 0.01; Conc.—concentration; DVF—desvenlafaxine; NPs—nanoparticles; adapted from Tong et al. [61], reproduced with permission from Elsevier [License Number 5426000412295].

**Figure 6 pharmaceutics-14-02742-f006:**
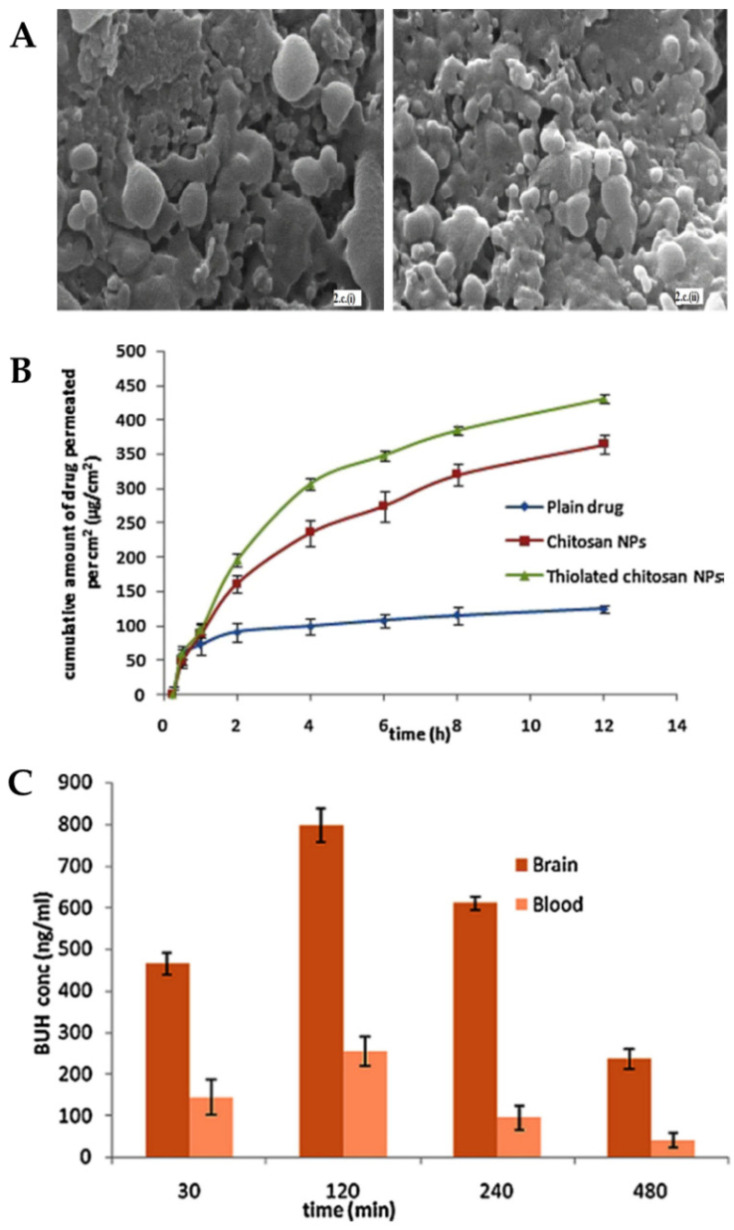
(**A**)—Scanning electron microscope images of the buspirone chitosan nanoparticles (on the left (2.c.(i))) and thiolated chitosan nanoparticles (on the right (2.c.(ii))); (**B**)—Ex vivo permeation study of buspirone chitosan nanoparticles (Chitosan NPs), thiolated chitosan nanoparticles (Thiolated chitosan NPs), and drug solution (Plain drug); (**C**)—Buspirone brain and blood concentrations after intranasal administration of the thiolated chitosan nanoparticles; BUH—buspirone; conc—concentration; NPs—nanoparticles; adapted from Bari et al. [64], reproduced with permission from Elsevier [License Number 5426000697685].

## Data Availability

Not applicable.

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
