# Peer review of "Lipid and Polymeric Nanoparticles: Successful Strategies for Nose-to-Brain Drug Delivery in the Treatment of Depression and Anxiety Disorders"

_pharmaceutics, 2022, doi:10.3390/pharmaceutics14122742_

Round 1

Reviewer 1 Report

Comments on pharmaceutics-2060961

In this manuscript, the authors reviewed the current progress for brain-targeted delivery of small molecular drugs using nanosystems via intranasal administration to treat depression and anxiety disorders. This is a good review with a clear and logical structure and the most recent publications in this research area. The authors are encouraged to address several issues to make this review more informative.

1. The common ligands that facilitate nanoparticle or drug transport to the brain via intranasal administration and their working mechanisms should be reviewed.

2. The authors need to provide a list of features or design criteria for nanosystems for brain delivery through this administration route.

3. The current limitations with reported nanosystems and what should be further investigated or developed should be discussed.

4. The biocompatibility of the reviewed nanosystems should be discussed as this is important for potential clinical translations.

5. Other than small molecule drugs, therapies using nucleic acids or proteins should be briefly discussed. Although this may not be the focus of this review manuscript, biopharmaceuticals may also provide viable treatments for depression and anxiety disorders, and thus they should be included in this review manuscript.

Author Response

We thank the reviewer for their helpful and insightful comments. The manuscript has been improved according to the reviewer’s suggestions (changes are highlighted in blue on the revised manuscript), and below is a point-by-point answer:

“1. The common ligands that facilitate nanoparticle or drug transport to the brain via intranasal administration and their working mechanisms should be reviewed.” – We thank the reviewer for their comment. A paragraph on the subject has been added on lines 858 to 868 accordingly.

“2. The authors need to provide a list of features or design criteria for nanosystems for brain delivery through this administration route.”  – We thank the reviewer for their remark. In fact, even though this information has been mentioned throughout the manuscript, an additional summary has been added on page 8 for a better understanding, in an additional figure, Figure 3.

“3. The current limitations with reported nanosystems and what should be further investigated or developed should be discussed.” – We thank the reviewer for their comment. This information has been added from lines 901 to 917.

“4. The biocompatibility of the reviewed nanosystems should be discussed as this is important for potential clinical translations.” - We thank the reviewer for their constructive criticism. This information has been added from lines 869 to 883.

“5. Other than small molecule drugs, therapies using nucleic acids or proteins should be briefly discussed. Although this may not be the focus of this review manuscript, biopharmaceuticals may also provide viable treatments for depression and anxiety disorders, and thus they should be included in this review manuscript.” – We thank the reviewer for their suggestion. A paragraph has been added on the subject, in lines 884 to 893.

Reviewer 2 Report

The Authors analyze the potential of intranasal nanoparticle administration for the treatment of depression and anxiety, using the analysis of previous studies. They concluded that the use of nanoparticles allows the drug´s protection from enzymatic degradation, and the modulation of its components allows controlled drug release and enhanced drug permeation. In vivo studies verified that the drug reachs the brain faster and leading to increased bioavailability and therapeutic effect with increased and prolonged brain drug concentrations.

The manuscript is suitable for publication in Pharmaceutiecs.

Author Response

We thank the reviewer for their comments, and for their favorable opinion on the publication of our manuscript.

Reviewer 3 Report

Margarida Alberto and colleagues review the current state of lipid and polymeric nanoparticles intranassally delivered in the treatment of depression and anxiety disorders. This is an interesting review to increase our knowledge on the use of different nanoparticles-biomaterials to sustain drug delivery in specific cerebral pathologies; the review is focused on depression and anxiety. In general the review is well written with extensive information on the material part. However, the medical/biological part tends to be weak. The manuscript has to be extended by adding details on the interaction between nanoparticles and the blood-brain-barrier, as well as information of nanoparticles kinetics. Also, the clinical context in relation with intranasal delivery of nanoparticles and drugs is marginally mentione, and has be improved. In several parts of the review, some criticism and the authors' own vision are missing.

Concerns:

1) In general it is confuse how nanoparticles cross the BBB and enter into the brain. A figure/scheme and a better detailed description should be provided to clarify the different BBB routes (transporter proteins, transcytosis processes, simple difusion, and paracellular transport). Please cite the most outstanding papers (no reviews) that have demonstrated these facts.

2) In the first part of manuscript, the authors mostly cite reviews instead of discuss and citing original sources/articles. The majority of references included in a review should be original articles; it sjould not be a review of reviews.

3) For how long nanoparticles remain in the brain? Do the nanoparticles enter inside the cell? Are the drugs coated in nanoparticles delivered intracelularly or extracellularly? Please, give detailed information and include specific original papers that cover these aspects

4) Please, define abbreviations. Explain the meaning of PDI and zeta potential for readers that are not familiar with these terms. How PDI, size and zeta potential influence intranasal drug absorption and Blood-Brain Barrier Penetrance?

5) Which nanoparticles formulations should be ideal for absorption and Blood-Brain Barrier Penetrance to enter the brain? Which nanoparticles show more life expectancy inside the brain parenchyma? I encourage the authors to include in the review tables that compare the advantages and disadvantages of different nanoparticles formulations, in general for cerebral pathologies (no only focused to depression and ansiety).

 5) Tolerability of nanoparticles in the clinical setting. Please discuss the past and current evidence of the use of nanoparticles administrated systemically or intranassally in humans for dianosis/treatment; independently of brain pathology. 

6) Proof of concept with the clinical use of drugs delivered intranasal to treat depression and anxiety disorders.

7)  How far we are towards the use of nanoparticles in the clinical setting (specially nanoparticles for brain disorders in general; and nanoparticles for depression and ansiety). In other words, is it the use of nanoparticles a preclinical epiphenomenon with complete lack of clinical translatability (not applicability at all) in the comming decades? Please discuss

Author Response

We thank the reviewer for the helpful and insightful comments. The manuscript has been improved according to the reviewer’s suggestions (changes are highlighted in blue on the revised manuscript), and below is a point-by-point answer:

“However, the medical/biological part tends to be weak.”

– Thank you very much for your opinion. We acknowledge that the information regarding medical/biological has not been greatly deepened, but this is due to the fact that the focus of the manuscript since its initial planning is, in fact, pharmaceutical technology. More specifically, the focus is the potential of intranasally delivered nanoparticles for the treatment of depression and anxiety disorders with emphasis on the formulation aspects. Nevertheless, we have made several additions regarding these topics to answer to the additional suggestions the reviewer made (see below).

“In several parts of the review, some criticism and the authors' own vision are missing.”

– Thank you very much for your valuable comment. The authors’ own critical perspective was included throughout the whole manuscript, including in section 2, where we not only state the results of the cited studies, but also provide our own critical opinion of the obtained results. Hence, it was not written as a mere “fact-stating”, but instead done with our own interpretation and critical perspective in mind. Additionally, when more than one study was carried out using the same drug molecule, at the end of the section there is a critical comment comparing the studies (for example in section 2.5, with the drug venlafaxine). Moreover, section 3, “Final remarks”, includes an additional critical perspective done by the authors, overall comparing the studies and concluding on their potential and potential improvements.

“In general it is confuse how nanoparticles cross the BBB and enter into the brain. A figure/scheme and a better detailed description should be provided to clarify the different BBB routes (transporter proteins, transcytosis processes, simple diffusion, and paracellular transport).”

– We thank the reviewer for their valuable comment.  This information has been added from lines 238 to 241, and 858 to 868.

“In the first part of manuscript, the authors mostly cite reviews instead of discuss and citing original sources/articles. The majority of references included in a review should be original articles; it should not be a review of reviews.”

– Thank you very much for your comment. We have added original research references whenever possible.

“Please, define abbreviations. Explain the meaning of PDI and zeta potential for readers that are not familiar with these terms”

- We thank the reviewer for the comment. A brief definition of PDI and zeta potential has been added on lines 341 and 342, accordingly.

“How PDI, size and zeta potential influence intranasal drug absorption and Blood-Brain Barrier Penetrance?”

– Thank you so much for the question. A paragraph has been added between lines 834 to 837 to properly answer it.

“Tolerability of nanoparticles in the clinical setting. Please discuss the past and current evidence of the use of nanoparticles administrated systemically or intranassally in humans for dianosis/treatment; independently of brain pathology.”

– We thank the reviewer for the very pertinent comment. We have added information on the subject from lines 869 to 883.

“Proof of concept with the clinical use of drugs delivered intranasal to treat depression and anxiety disorders.”

– Thank you so much for your comment. Clinical evidence has only surfaced in recent years, information on the subject has been added on lines 894 to 900.

“How far we are towards the use of nanoparticles in the clinical setting (specially nanoparticles for brain disorders in general; and nanoparticles for depression and ansiety). In other words, is it the use of nanoparticles a preclinical epiphenomenon with complete lack of clinical translatability (not applicability at all) in the coming decades? Please discuss”

– Thank you very much for your constructive question. A discussion regarding this relevant issue has been added on lines 884 to 893, and 901 to 917.

Round 2

Reviewer 3 Report

The authors have anwered most of the concerns and discussed the clinical context. Changes have improved the manuscript